# Gender and Age Effects on Public Attitudes to, and Knowledge of, Animal Welfare in China

**DOI:** 10.3390/ani12111367

**Published:** 2022-05-26

**Authors:** Francesca Carnovale, Jin Xiao, Binlin Shi, David Arney, Kris Descovich, Clive J. C. Phillips

**Affiliations:** 1College of Animal Science, Inner Mongolia Agricultural University, Zhaowuda Road 306, Hohhot 010018, China; francesca.carnovale@student.emu.ee (F.C.); shibinlin@yeah.net (B.S.); 2Chair of Animal Nutrition, Institute of Veterinary Medicine and Animal Sciences, Estonian University of Life Sciences, Kreutzwaldi 1, 51014 Tartu, Estonia; david.arney@emu.ee (D.A.); clive.phillips@curtin.edu.au (C.J.C.P.); 3School of Veterinary Science, The University of Queensland, Gatton, QLD 4343, Australia; k.descovich1@uq.edu.au; 4Curtin University Sustainability Policy (CUSP) Institute, Curtin University, Perth, WA 6845, Australia

**Keywords:** animals, animal welfare, China, attitudes, knowledge, livestock, management, gender

## Abstract

**Simple Summary:**

Gender has been found to influence attitudes towards animals, with women demonstrating more positive attitudes than men in some countries. As attitudes determine consumer behaviour, to a certain extent, and China (the biggest livestock producer globally) has witnessed major social changes in recent decades, we conducted a survey to investigate whether gender and age influenced attitudes towards animals. Respondents self-classified their gender as female, male, other, or they did not disclose it. We found that the attitudes were determined by a combination of gender and age, with more support for animal welfare in women aged 18–24 years than in older men (25–54 years). Those that did not disclose their gender and those declaring it as ‘other’ appeared to have different attitudes to those declaring it as female or male.

**Abstract:**

A person’s gender and age can influence their attitudes towards animal welfare, with more benign attitudes generally ascribed to women. Given that attitudes influence consumer behaviour and the rapid recent social development in China (globally the biggest livestock producer), we surveyed over 1300 individuals across China to elucidate the role of gender and age in determining attitudes towards animals. Respondents self-identified their gender as male, female, other or not revealed. There were interactions between age and gender for many of the survey items, demonstrating that the effects of gender were dependent on the respondents’ age. Women aged 18–24 reported more benign attitudes towards animals than older men (aged between 25 and 54 years, depending on the survey question) and more empathetic responses were found in young respondents generally, although this did not necessarily translate into a willingness to pay more for higher-welfare animal products. We propose, drawing on Social Identity Theory, that women see animals as part of their social group, whereas men tend not to do this. Those responding as neither male nor female, i.e., as another gender, and those not revealing their gender appeared to have different relationships to animals than those responding as men or women. It is concluded that within Chinese culture, attitudes towards animals and their welfare are complex and influenced by an interaction between gender and age.

## 1. Introduction

Large differences in attitudes towards animals have been recorded, arising from characteristics such as gender, age, education level, field of work, religion, culture, and nationality [1,2,3]. In relation to gender disparities in attitudes towards animals, Pifer in 1994 [4] first reported that women disagreed more than men with the scientific use of animals, believing that it causes pain and injury to animals. Additionally, animal rights activists are more commonly women [5]. In other research, women expressed more opposition than men towards scientific experimentation using animals and hunting, while men had more concern for the preservation of wildlife habitats [6,7].

Gender differences in attitudes towards animal welfare topics have been demonstrated across several different countries and cultures. In the USA, women are more concerned with animal protection than men are [8]. Similarly, in the Eastern European countries of the Czech Republic, Slovakia, Croatia, Moldova, Slovakia, and Ukraine, women were found to have a greater propensity to be concerned about animal welfare, compared to men [9,10]. Additionally, female, but not male, Portuguese students were found to consider killing animals for human food consumption not an acceptable reason for the slaughter of animals [11]. In contrast, however, previous studies in the People’s Republic of China (hereafter China) [12,13] found similarities between women and men regarding their attitudes towards animals. However, Li et al. [12] focused only on the transport and slaughter of animals, and Su and Martens [13] reported that women and men answered in a similar way to general questions regarding their attitudes towards animals.

Global differences in the attitudes of women and men towards animals may be explained by a variety of factors, such as political and cultural influences and the level of dependence of women on men. In a cross-country study across Eurasia, women only tended to report greater concerns for animal welfare than men in countries with a high level of gender equality [14]. This has also been demonstrated at a household level; women with high social, economic, and interpersonal dependence on their husband had attitudes towards animals that were more similar to those of their male partners [15].

Gender differences in attitudes and behaviours toward animals are not immutable. They tend to develop during or even before adolescence [7,16]. In particular, experience of companion animals during childhood increases concern for the welfare of non-human animals in adulthood [17,18]. This relationship has been confirmed in several culturally diverse populations such as British and Japanese [19].

Age can also be an important factor that influences attitudes towards animals. In their review, Ormandy and Schuppli [20] noted that, in the USA, people under 35 years of age tended to have more positive attitudes towards animals than older people (over 56), who held more utilitarian attitudes. Similarly, US adolescents aged between 14 and 19 thought it was less acceptable to use animals as research subjects than older individuals [21]. This may relate to the extent of contact with animals. Older persons, in particular those over the age of 65 years, are less likely to own a pet than younger individuals, and this may reduce their engagement and empathy with animals [22]. Much of this previous research on attitudes of Chinese people focused on students as the respondents. However, it is important to further understand the attitudes of society at large, including across all age groups. Knowledge about the differences between age groups could be useful when endeavouring to improve animal welfare, as increased attention could be given towards those age groups displaying less empathy towards animals.

These differences in attitude resulting from age are evident in diverse cultures. Young people in China have more positive opinions towards animals than middle-aged and older adults [13], in that they are less likely to accept “animal use”, “destruction of animal integrity”, “killing of animals”, “testing on animals” or “harm to animals for the benefit of the environment”. Some researchers have sought to understand the age differences by analysing the link between ethical ideologies and attitudes toward animals [23,24]. There are also age-related differences in knowledge, including knowledge of societal attitudes toward animals. For example, Davey, 2006, suggests that young people in China have tended to have better educational opportunities, including different cultural education, and have gained more knowledge relating to animal welfare than older people [25]. Although animal welfare is still an emerging concept in China, younger generations are generally more aware of it than older generations and show more positive attitudes toward animals [26].

In a recent study, Carnovale et al., (2020) [27] identified and described attitudes towards, and knowledge of, animal welfare in the Chinese public. Although most of the public in China were unfamiliar with the term “animal welfare”, respondents indicated significant concern and care for animals, especially for wild animals. Although reported attitudes do not always align with behavioural intentions, most Chinese people surveyed indicated a willingness to pay more for food products derived from animals raised in good welfare conditions, and they believed that animal welfare can have an impact on food safety and product quality. Most respondents also believed that there should be legislation that protects the welfare of animals, for example to avoid long transportation times and to implement stunning before slaughter.

In the current study, data from the same questionnaire as used by Carnovale et al. [27] were used to explore differences in the opinions and attitudes of the Chinese public towards animal welfare arising from their age and gender. It was hypothesised that, among the general public in China, attitudes towards animals would differ with gender, and that attitudes towards animals would differ with age.

## 2. Materials and Methods

The questionnaire and survey method were approved by the Human Research Ethics Committee of the University of Queensland, Australia (#2019001811) and have been previously described in full [27] (Appendix A, Box 0). In brief, the survey was administered by undergraduate students from the Inner Mongolia Agricultural University, with a total of 2170 questionnaires distributed between August 2019 and August 2020. Questionnaires were delivered in all 23 of the directly administered provinces of the People’s Republic of China. Potential respondents were individually approached in public spaces (e.g., shopping centres, streets, parks, squares, markets) and by door-to-door knocking at residences. The survey’s format and content were initially developed in English before being translated into written Chinese (Zhongwen) by bilingual collaborators from the Inner Mongolia Agricultural University, who were proficient in Chinese and with the animal welfare terminology used. The Chinese version was then back translated into English and compared with the original version to ensure the original meanings were retained. Section 1 of the questionnaire focused on demographics: age, gender, level of education, professional background, religious affiliation, and place of residence. This section also asked about the participant’s background and knowledge concerning animal welfare (e.g., ‘Where did you learn about caring for animals?’ and ‘Who do you think is most responsible for the adequate care of animals?’. Section 2 was structured into four sets of questions asking which groups of animals they cared about most, with answers selected from 5-point ordinal scales. Lastly, participants were asked for their reasons why they believed that animals should be cared for, and which aspects of welfare they believed were most important.

The current paper focuses on the effect of two of the most significant demographic factors—age and gender—which are recognised to be influential on attitudes towards animals. The effects of further demographic variables will be described in subsequent papers. Survey respondents were asked to self-identify their gender by selecting one of four categories: Male, Female, Other, and ‘I prefer not to say’ (Non-revealers). We used the term ‘gender’ here, rather than ‘sex’, because the former refers to social and cultural differences in identity, expression, and experience, rather than sex, which is the biological characteristic recorded at birth (Ontario Human Rights Commission, 2021) [28]. We did not specifically define the term gender in the survey; therefore, selection was determined by the participant’s own understanding of the categorisations. “Men” and “women” were therefore defined as persons who described their gender as male and female, respectively. “Other” was the term used for identified persons who reported their gender to be other than male or female, and is separate from those who preferred not to disclose their gender (Non-revealers). Age was recorded as an ordinal age category with options being 18–24; 25–34; 35–44, 45–54; 55–64; and 65 and over, indicating their age in years at the time of the survey.

Response variables that were structured as ordinal categories were transformed to a numerical scale from 1 to 5 in order, from negative to positive affirmation, for example, 1 (not sure) to 4 (many times); 1 (not at all) to 5 (to a great extent); 1 (not at all) to 5 (extremely); 1 (definitely not) to 5 (definitely); 1 (very poor) to 5 (very good); 1 (much worse) to 5 (much better); 1 (strongly disagree) to 5 (strongly agree); and 1 (not at all important) to 5 (very important). A six point scale (1 (5%) to 6 (>100%)) was used to ask how much extra people would be prepared to pay for high welfare. In the case of binomial categories, the data were transformed into “1” for the higher number of responses and “2” for the lower number of responses.

### Statistical Analysis

All analyses were conducted using the statistical package Minitab 18.0 (Minitab Version 18; Minitab Inc., State College, PA, USA). Descriptive statistics were generated and have been previously reported [27]. The effects of non-demographic questions in the survey were analysed by Ordinal Logistic Regression for ordered categorical dependent variables, in order to predict the effects of demographic explanatory variables (including gender and age). Where interactions were significant (*p* < 0.05), results for a model including the relevant interactions are provided; otherwise, results for the relevant models without interactions are provided. For binary and nominal dependent variables, Binary and Nominal Logistic Regression models, respectively, were used. Data are displayed as the mean response on the relevant ordinal scale and the number of responses for each group (Gender: Male, Female, Other (Non-binary), Non-revealers; and Age: 18–24, 25–34, etc.) as relevant. As some gender categories had only a small number of responses, they were not able to be included in the interaction models. These were the non-binary gender category and the two oldest age groups, 55–64 and ≥65 years.

## 3. Results

In total, 2170 people were approached to participate and, of these, 1301 completed the questionnaire. Overall responses have been reported previously in Carnovale et al. [27]; however, two important and interacting demographic influences were detected in the logistic regression analyses: gender and age. This paper focuses on these effects.

Respondents almost equally self-reported as male or female: 47% and 48% (n = 618 and n = 628, respectively). Of the remaining participants, 3% (n = 39) identified as Other (non-binary) and 0.5% (n = 7) said that they preferred not to reveal their gender (non-revealers). For most gender categories, participants were most commonly between 18 and 24 years of age (30% of women; 37% of men; 29% of non-binary participants; and 41% of non-revealers) (Figure 1). For the 25–34 age group, there were 7% fewer men than women. For the age categories of 35–44 and 45–54, the percentages of men and women were similar. For participants 45–54, 55–64, and ≥64 years of age (5% of the total pool of respondents), only one respondent per age category identified as non-binary (other). For brevity, only significant relationships are reported. If response variables are not included, it should be understood that there were no significant differences. A complete table with non-significant results is included in Appendix A, Table A1 and Table A2.

### 3.1. Interactions between Age and Gender on Animal Welfare Opinions

Women that were younger did advance more pro-animal welfare opinions than the other groups. In particular, women aged 18–24 (n = 189) were more likely to say that they lived in harmony with animals compared to men aged 35–44 years (n = 114) or non-revealers aged 25–34 years (n = 10) (Table 1). They were also more likely to say that caring for animals is important compared to men in the 25–34 year (n = 139) and 45–54 year (n = 88) age groups. Middle-aged men (35–44 years and 45–54 years) were more likely than young women (18–24 years) to say that the standard of animal care in China was better than in other countries, and less likely to agree that a comfortable environment was important for animal care (35–44 age group only). Young women (18–24 years) were less likely than most older male age groups to agree that it is OK to buy products of animals that have suffered if the product quality is good enough or if the price is low enough (Table 1).

Compared to young women (18–24 years), non-revealers (35–54 years; n = 6) were more likely to say that animal welfare was worse in China compared with other countries, although the sample size was small. Both groups were generally in agreement about the importance of caring for a range of different animals. Importance scores were significantly higher for the group of young women than the non-revealers, for the animal categories of mammals, reptiles, birds, and stray animals (Table 1); however, these differences were not uniform across age groups. Non-revealers aged 25–34 years were less likely than young women (18–24 years) to agree that animals should be cared for in order to improve product quality or taste.

### 3.2. Gender Effects on Attitudes of Chinese Respondents towards Animal Welfare

In comparison to either men or women, non-revealers indicated they had learned about animal care mainly from family and friends, more commonly than either men or women (Table 2). Of non-revealers, 60% were prepared to pay more (female 61% and male 56%). Most of the men (60%) and women (65%) were prepared to pay at least 10%, while for non-revealers, most (70%) were prepared to pay 20% more for a product from an animal that had been very well cared for.

Participants who were non-revealers reported the standard of animal care in China as poorer than the other gender categories did, and they were more likely to report it as being worse than other countries (Table 2). In terms of the importance of caring for different animal groups, there were differences between people who identified as non-binary and the remaining gender categories, although the number of such respondents is recognised as being too small for definitive comment. When asked how important it was to care for reptiles, insects, experimental animals, agricultural animals, stray animals and wildlife, on average this group gave neutral to slightly positive scores, which were lower (indicating less importance) than those given by men, women, or non-revealers (Table 2).

Participants were asked how strongly they agree with a series of statements on why animals should be cared for (e.g., “*It makes me feel good*”, “*To improve profit from animals*”) with a higher score indicating stronger agreement. There were few differences between gender categories for these statements, with the exception of “*For food safety*” where non-binary participants reported a lower agreement with this statement than other groups, and “*For the sake of the animals*” where the highest importance score was given by men. Gender differences were also apparent in attributing importance to specific conditions provided for animals, namely “*A comfortable environment*”, “*Opportunity to perform natural behaviours*”, “*Absence of fear or distress*”, and “*Absence of pain*”. Participants from all gender categories gave positive scores for these on average, indicating a belief that these are important; however, scores from non-binary participants were typically lower than for the other groups (Table 2).

Gender differences were also found in the level of acceptance that participants had for different invasive procedures that are commonly performed on animals for management purposes (e.g., ear tagging, castration and tail docking). People from all gender categories were largely supportive of these measures with mean scores of 3.29 or higher, on a 5-point scale from 1 (Strongly disagree) to 5 (Strongly agree). However, these were supported the least by women (Table 2). Participants from all gender groups indicated, on average, that legislation is important to ensure adequate animal care, although differences were observed between groups with non-binary participants reporting a lower importance score than men, women or non-revealers (Table 2).

Most of the respondents of all the gender groups declared that they had never heard of animal welfare, but they thought it was extremely essential to care for animals, and perhaps having the possibility to learn about this topic at school would be beneficial. All gender categories felt society was responsible for animal welfare. Pets, mammals, and birds were the animal taxa on which all genders agreed on their importance. For the sake of the environment and human health, the majority of all gender categories considered it important to care for animals. All respondents indicated they care about animals for religious reasons and to make themselves feel better. Increasing profits and increasing the quality of animal products were similarly important to the respondents, regardless of their gender. Irrespective of the gender of the respondents, it was thought to be important that animals be able to exercise for physical fitness and have control over the environment in which they live. It was also thought to be crucial to have enough area and access to drinkable water and prevent contracting diseases or injury. The management of animals was declared important by the majority of all gender categories, especially concerning transportation time and slaughter management, and they all thought it extremely important to have an animal protection organisation.

### 3.3. Age Effects on Attitudes of Chinese Respondents towards Animal Welfare

Many age-related differences were found in participant attitudes towards animals. The youngest respondents (aged 18–24 years) were more likely to have heard of the phrase “animal welfare” than those aged 65 years or older. They were also more likely to report that they lived in harmony with animals, and that caring for animals was important to them as a person, in comparison to several of the older age groups (25–34, 35–44, 45–54) (Table 3) but not compared to those older than 55. In contrast, the youngest respondents were less supportive of animal care being taught in schools than the oldest respondents (65 and older) (Table 3). In terms of whether they were willing to pay more for products from animals that were better cared for, the younger respondents were less likely to pay more than respondents aged 45–64, with those 65 years and older also giving this less support (Table 3).

The youngest respondents were more likely to believe that all of society is responsible for animal care than those aged 45–54 years (Table 3). They were also more likely to attribute importance to caring for reptiles, insects and pet animals compared with those aged 35–44 and, in the case of pet animals, those aged 45–54.

Age effects were evident for participants’ views on why animals should be cared for, and while these effects were mixed, in general, younger respondents were slightly less likely to agree that animals should be cared for because it made them feel good, for the sake of the animals, for religious reasons, or to improve profit, than those in older age brackets (Table 3). The youngest respondents were more likely to believe that nutrition, control over their environment and absence of pain were important for animal care than those who were slightly older, in the 25–34 age bracket, although on average both groups assigned high levels of importance to these conditions. The oldest respondents believed more than the other age groups that animals on farms should have enjoyable experiences, while the youngest participants were less likely than the 25–34-year-olds to agree that it is OK to buy products of animals that have suffered if the price is low enough (Table 3).

Some factors were consistent across age groups. The age of respondents had no effect on their judgment of animal welfare in China as “poor” and, compared to other countries, they agreed that it was “somewhat worse”. In all taxa animals listed in this survey, mammals, birds, and stray animals were identified by all respondents of any age as important, while wild animals, experimental and agricultural animals were identified as extremely important. Regardless of age, all respondents agreed that the most important reasons for caring for animals were food safety, human health, improving the quality of animal products, and the need for animals to be free of fear and allowed to exhibit normal behaviour. The reasons given for the care for animals are that the animals need to have the possibility for physical fitness and control over their environment. Animal care and management was found to be very important regardless of respondent ages, particularly in terms of transit time and slaughter management, but it was also thought to be highly necessary to have an animal protection organisation and law that protects animals.

## 4. Discussion

### 4.1. General Effects of Gender and Age on Attitudes to Animals

Previous surveys, mostly based in Western countries, have identified differences between men and women in their opinions concerning animal welfare [4,14,15,22]. Women commonly, although not universally, express greater concern for animals than men [29]. However, gender-based differences towards animals are not consistent across geographic areas [14], thus findings from one region or country are not necessarily representative of others. The current study, based in China, identified several gender-related differences in attitudes towards animals, as well as their care and welfare. This differs from previously published literature, for example research based in China by Li et al., 2018 [12] and a study by Erian et al., involving a range of Asian countries [30], where no differences were found.

In general, respondents who were non-revealers (n = 39) expressed greater sensitivity to animal welfare on most questions, and their views differed from both men and women. Attitudinal differences between men and women have been reported across many contexts in relation to animal welfare [2,3,4,5,6,7,8,9], such as animal research use, companion animals, and animal protection and rights. These differences likely arise from a complex interaction of the person’s demographics, e.g., gender and age, and their sociocultural experiences. Gender has been used as the descriptor throughout this paper. As described by the American Psychological Association, gender refers to the attitudes, feelings, and behaviours that a given culture associates with a person’s biological sex [31].

In the current study, gender was not limited to male and female, but also included options for participants to self-identify as a different gender (non-binary, n = 7) and to refrain from disclosing their gender (non-revealers). In the past, surveys have often used exclusively binary gender categories, which do not allow adequate representation across this influential demographic variable. Methodologically, and ethically, surveys should include multiple response options for gender [32] as gender identity is complex and can be fluid, and the inclusion of sufficient response options enables both representation of participants’ identities and more reliable survey results [32]. While the number of non-binary-declaring respondents was small, they are included as their responses are entirely novel and, despite their small number, suggest that there may be differences in attitudes to the mainstream male and female identities.

Participants in the current study of different ages and genders were asked to what extent they live in harmony with animals. To live in harmony with animals and nature is an important traditional principle in China that predates even Confucianism [33]. It is exemplified by the “Great spiritual transformation” dictated by Confucianism, that the evolution or transformation of nature and the actions of humans must be in equilibrium [34]. According to Confucian teachings, animals are considered to be an integral part of the productivity, richness, diversity, and beauty that are central features of the concepts of Heaven and Earth, and considerations for well-being must also encompass the well-being of non-human beings and natural processes [35]. In our study, the youngest respondents (age 18–24 years) indicated that they live harmoniously with animals more than older respondents (25–54 years), although there was no difference from those in the two oldest age groups (55–64, and 65 and older), and responses were also influenced by an interaction between gender and age. In general, the youngest respondents demonstrated more pro-welfare attitudes than older age groups, although the differences were largely between those in their late 20s and those in their early 50s. In 2018, the Chinese government introduced non-mandatory animal welfare classes to be delivered in high schools [36,37]. This move aligns with the views of respondents in all age groups from this survey, who were in favour of animal care being taught in schools, although the oldest age group actually demonstrated more support than the youngest group. The more pro-animal sentiments found in the youngest respondents in our study may be attributed to several possibilities. Firstly, it may be due to the emergence of animal welfare terminology relatively recently in China, allowing younger respondents to have more familiarity with these issues [38]. It is also possible that older respondents may be more likely to prioritise human welfare, their own or others, above animals [22]. Another possible reason is that older respondents may have less contact with companion animals than those who are younger [22]. Contact with companion animals, particularly at a young age, has been shown to positively influence attitudes towards animals and their welfare [18,39]. In China, recent prosperity, especially in the working sectors of society, has only now made pet ownership possible for most people [40], which may mean that younger respondents have had more opportunities for pet ownership or companion animal contact. The youngest respondents typically gave higher importance scores than older participants about the need to care for reptiles, insects and pet animals, and also indicated a stronger belief that all of society is responsible for animal care, although again the oldest participants showed more alignment with the youngest, than those in the middle age groups. Middle-aged respondents (from 25 to 54) were more likely to agree that people care for animals for religious reasons, possibly reflecting the Buddhist and Taoist traditions in this country, which both revere living things [41,42]. All participant age groups indicated that animals on farms should be provided with enjoyable experiences; however, the oldest age group was more supportive of this statement than the youngest group. It is possible that some age-related differences are due to increased urbanisation in China [43]. Urbanisation has been documented in several countries, including China, to diminish the relationships that people have with nature and animals [44], and where this is the case, it is likely to affect age groups disproportionately. As China is transitioning away from its largely agrarian economy, older respondents are more likely to have had experience of tending animals on farms.

Younger respondents (18–24 and 25–34 years) and those in the oldest group showed the most reluctance to pay more for products from animals that were better cared for. This may be reflective of their relationships with animals but is also likely to be influenced by their financial resources or purchasing patterns. Household income is known to affect purchasing decisions in China for other ‘premium’ products such as organic food, although these issues interact as animal welfare is also a motivator for the purchase of organic products in China [45]. Generational differences in attitudes towards animals have been found in previous research [46], in relation to nature and wild animals. In that research, older generations typically had stronger views about the protection of wild places and wild animals, while younger generations showed the most concerned for animals and the most interest in participating in direct animal experiences such as pet keeping [47]. Along with our own study, this suggests that attitudes towards animals are multidimensional and positive alignment with one dimension may differ in another, with attitudinal shifts occurring from one generation to the next in complex ways.

### 4.2. Interactions between Age and Gender on Attitudes to Animal Welfare

In the current study, middle-aged male participants typically had less positive attitudes towards animals than younger female participants. This gender difference aligns with previous research conducted internationally that also found more benevolent attitudes in women towards animals, particularly in regions with a high level of gender empowerment [14]. However, in that study, no gender difference was found for China at that time (data collection was in 2007–2008). This may be due to differences in sampling, as the current study had a sampling skew towards the province of Inner Mongolia, which has implications because of the ethnic diversity within China and regional differences in the relationships between people and the natural environment, e.g., [48,49], or may be due to fluctuating gender roles and gender attitudes in China. In China, as in most other countries, gender gaps in societal roles and equality of opportunities are evident. Women are paid around three quarters of the salary paid to men [50], and economic reforms, such as those in the late 1990s and early 2000s, have disproportionately affected the participation of women in the workforce [51]. Women tend to bear a higher burden of household duties, including childcare, which also affects their employment prospects as well as earning potential, and influences attitudinal bias in employers [51,52]. They are also more likely to be responsible for food purchase and preparation, which may again influence their attitudes towards animal welfare, as many participants in this research drew associations between animal welfare and product safety [52]. Previous research from Canada found strong gender effects on solidarity of participants with animals [53] and also found negative associations with sexist attitudes, although in that study no gender–age interaction was found. While the same may not necessarily be true for participants in China, the role of animals in Chinese society is changing, notably for women [54], and drawing on social identity theory [55], our results suggest that young women may view animals as within their core social group, more than middle-aged males do. If young women experience a solidarity with animals that older men do not [53], this may comparatively reduce their prejudice against animals, resulting in more benevolent attitudes towards animal welfare and management.

### 4.3. Non-Revealers and Non-Binary Participants

In our study, only a small number of participants chose not to reveal their gender; however, those in this category expressed high levels of sensitivity to animal welfare for many survey items, and on several aspects their views differed from the other gender groups. Non-revealers were more likely to say that they learned about animal care from family and friends, but no difference was found for other sources of information, such as formal study. They were more likely than the other gender groups to say they lived in harmony with animals. In contrast, they were less supportive of looking after animals for the animal’s own sake than other groups, and less supportive of doing so for food safety reasons than men or women.

Failure to reveal gender may occur for several reasons. In our case, non-revealers may perceive themselves to fall outside of the conventional binary assignation of gender roles. Previous research suggests that non-revealers to gender identity questions can occur for several reasons, including participants being unsure of their own gender at the time, undergoing gender transitioning, or discomfort with the other options given [56]. However, in many respects, this group differed in their attitudes towards animals from those who identified their gender as being non-binary, which suggests non-disclosure could be for other reasons, such as the perception of risk around privacy when providing personal information. This cannot be determined from the current results; however, previous literature has shown that individual variation in risk tolerance and behaviour is consistent across diverse contexts [57] and may therefore also potentially influence attitudes towards animals, or the assessment of risks/benefits around their management.

We anticipated that non-binary participants in this study would demonstrate more benevolent attitudes towards animals, as previous research suggests marginalised members of society often develop strong attachments to companion animals, e.g., [58,59,60], and people in the traditional conservative society who are gender non-conforming commonly experience social stigma [61]. The current results contradicted these expectations. Compared to the other gender groups, non-binary participants gave lower scores for living in harmony with animals and were more likely to indicate that the standard of animal care in China was higher than other countries. They also attributed a lower amount of importance towards the care of most animal groups, with the exception of mammals, birds and pets. Similarly, when asked to rate the importance of specific living conditions for animals, they gave lower scores for several attributes including the provision of a comfortable environment, opportunity to perform natural behaviours, absence of fear/distress, and absence of pain.

### 4.4. Limitations of the Study

The number of non-binary respondents in the current survey was very low in comparison to those who identified as male or female; therefore, these results should be considered preliminary until they are more systematically explored in future research. Imbalance in participant numbers for gender and age is a limitation of this research; however, this study provides important insight into the influence of these demographic variables on attitudes towards animals in China.

Another potential limitation of this work is that in mainland China, the English words “gender” and “sex” are translated to the same word (性别) and official documents generally only allow for two categories (male or female), which may have influenced how participants responded to this question in the survey. Some misunderstanding of the translated version of the questionnaire may have occurred, which could be a limitation, although this was not reported by the questioners.

### 4.5. Practical Implications of the Results

This study provides evidence that the attitudes of the Chinese population, and relationships between these attitudes and age and gender, are not greatly different from those recorded elsewhere; the sensitivity of the Chinese population to animal welfare does not differ much from the rest of the world. The identification of gender and age groups that are more concerned with the welfare of animals confirms the value of, and interest in, educating the young in the formal school setting on issues of animal welfare. Whether this interest is related to the age of the respondents, where we might expect that their attitudes would harden as they get older, or if it is a generational factor, with continued interest into later life, could be confirmed by a longitudinal study, ideally asking the same respondents in later life, but this is beyond the bounds of this study.

## 5. Conclusions

This study investigated the effect of gender, age, and their interactions on Chinese respondents’ attitudes to animals and their welfare. While age was influential on survey responses, the effect was not linear. Young women tended to have more empathetic attitudes towards animals than men aged between 25 and 54 years. Similarly, young participants had more benevolent attitudes towards animals than middle-aged participants; however, they aligned in many aspects with the oldest age groups. Gender also influenced survey responses, with non-binary participants generally indicating less positive attitudes towards animals and their management, and those who chose not to reveal their gender indicating more positive attitudes. However, these results should be considered preliminary as only a small proportion of respondents chose gender categories other than male or female.

## Figures and Tables

**Figure 1 animals-12-01367-f001:**
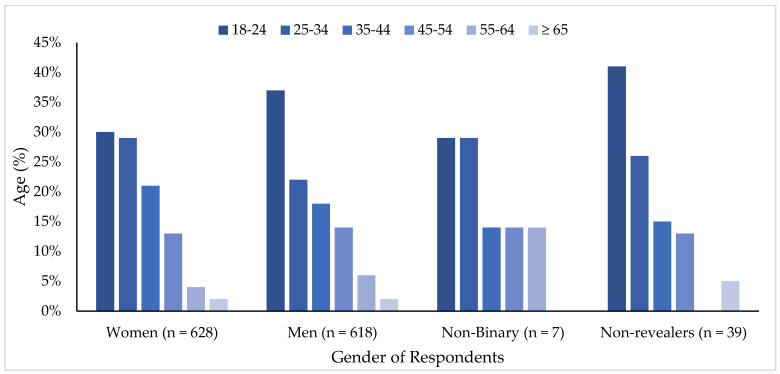
Percentage of respondents in each age category within self-identified gender groups (n = 1301).

**Table 1 animals-12-01367-t001:** Significant (*p* < 0.05) interaction effects between gender and age on attitudes towards animal welfare in China. Mean responses on 5-point ordinal scales are provided. For gender categories, the number of respondents was 618 men; 628 women; 39 non-binary participants. For age categories, the number of respondents was 430 18–24 years; 337 25–34 years; 252 35–44 years; 176 45–54 years.

Questions and Responses	Women Aged 18–24 Mean Score	Comparison Gender × Age Groups	Mean Scores	Odds Ratio	Coef.	95% CIs Lower Upper		*p*-Value
Do you live in harmony with animals? 1 (Not at all) to 5 (Extremely) ^1^	3.41	Men × 35–44	3.21	0.48	−0.73	0.26	0.87	0.013
		Non-revealers × 25–34	2.95	5.79	1.75	1.12	30.03	0.036
How important is caring for animals to you as a person? 1 (Not at all) to 5 (Extremely) ^1^	3.58	Men × 25–34	3.28	0.43	−0.83	0.25	0.76	0.003
		Men × 45–54	3.27	0.48	−0.74	0.24	0.94	0.034
Do you think that animal care should be taught in schools? 1 (Definitely not) to 5 (Definitely) ^2^	3.26	Men × 25–34	3.39	0.56	−0.57	0.32	0.98	0.042
How do you think the standard of animal care in China compares to other countries? 1 (Much worse) to 5 (Much Better) ^3^	2.33	Men × 35–44	2.48	3.09	1.12	1.68	5.68	<0.0001
		Men × 45–54	2.45	2.14	0.75	1.07	4.26	0.031
		Non-revealers × 35–44	1.93	7.38	1.99	1.02	53.66	0.048
How important is it that the following animals are cared for? 1 (Not at all important) to 5 (Very important) ^4^ Mammals	4.00	Non-revealers × 35–44	2.64	14.96	2.70	2.34	95.60	0.004
Reptiles	3.85	Non-revealers × 35–44	2.67	8.38	2.12	1.32	53.06	0.024
Birds	4.01	Men × 45–54	4.09	0.43	−0.84	0.21	0.87	0.021
		Non-revealers × 35–44	3.16	6.27	1.90	1.02	44.14	0.047
Pet animals	4.14	Non-revealers × 25–34	3.62	7.66	2.03	1.37	42.78	0.021
Stray animals	4.12	Non-revealers × 35–44	3.20	10.05	2.30	1.53	66.19	0.016
Why care for animals? Indicate how strongly you agree or disagree with the following reasons: 1 (Strongly disagree) to 5 (Strongly agree) ^5^ To improve product quality or taste	3.88	Non-revealers × 25–34	3.67	10.45	2.34	2.00	54.66	0.005
How important are the following conditions in animal care? Indicate how strongly you agree or disagree with the following reasons 1 (Not at all important) to 5 (Very important) ^4^ A comfortable environment.	4.33	Men × 35–44	4.19	0.42	−0.87	0.22	0.79	0.008
Indicate your level of agreement with the following statements: 1 (Strongly disagree) to 5 (Strongly agree) ^5^ Farms with animals should be certified by animal protection organizations.	3.91	Men × 25–34	3.99	0.45	−0.79	0.25	0.81	0.008
It is OK to buy products of animals that have suffered if the product quality is good enough	3.29	Men × 35–44	3.56	2.76	1.01	1.53	5.00	0.001
		Men × 45–54	3.38	2.62	0.96	1.34	5.14	0.005
It is OK to buy products of animals that have suffered if the price is low enough	3.01	Men × 25–34 Men × 35–44 Men × 45–54	3.40 3.43 3.24	2.20 3.84 3.70	0.78 1.34 1.30	2.20 3.84 3.70	3.80 6.95 7.26	0.004 <0.0001 <0.0001

^1^: 1 = Not at all, 2 = Slightly, 3 = Moderately, 4 = Very much, 5 = To a great extent/Extremely. ^2^: 1 = Definitely not, 2 = Probably not, 3 = Possibly, 4 = Probably, 5 = Definitely. ^3^: 1 = Much worse, 2 = Somewhat worse, 3 = About the same, 4 = Better, 5 = Much Better. ^4^: 1 = Not at all important, 2 = Slightly important, 3 = Neither important nor unimportant, 4 = Somewhat important, 5 = Very important. ^5^: 1 = Strongly disagree, 2 = Disagree, 3 = Neither agree nor disagree, 4 = Agree, 5 = Strongly agree. Coef. = coefficient.

**Table 2 animals-12-01367-t002:** Gender effects on respondents’ attitudes towards animal welfare in China. Mean responses on 5-point ordinal scales are provided. Differences are presented between Men (n = 618), Women (n = 628), Non-binary participants (n = 7) and Non-revealers (n = 39).

Questions and Responses	Men	Women	Non-Binary	Non-Revealers	Odds Ratio	Coef.	95% CIs Lower Upper		*p*-Value
Do you live in harmony with animals? 1 (Not at all) to 5 (Extremely) ^1^	3.32	3.33	3.00	3.69	0.47	−0.76	0.24	0.90	0.023
Where did you learn about caring for animals from? Family and friends (1 = yes, 2 = no)?	1.36	1.32	1.42	1.48	1.35	0.29	1.12	1.61	0.003
How much more would you be willing to pay for a product from an animal very well cared for compared with the standard product? 1 (5%) to 6 (>100%) ^2^	2.24	2.26	3.16	3.03	0.37	−2.83	0.19	0.74	0.005
What do you think is the current standard of animal care in China? 1 (Very poor) to 5 (Very good) ^3^	2.58	2.53	2.50	2.18	2.19	0.78	1.11	4.32	0.024
How do you think the standard of animal care in China compares to other countries? 1 (Much worse) to 5 (Much Better) ^4^	2.42	2.31	3.42	1.97	0.20	−1.58	0.04	0.93	0.039
How important is it that the following animals are cared for? 1 (Not at all important) to 5 (Very important) ^5^ Reptiles	4.01	4.01	3.14	3.94	4.92	1.59	1.08	22.9	0.039
Insects	3.79	3.91	3.28	3.64	1.39	0.32	1.11	1.73	0.004
Experimental animals	4.19	4.25	3.28	4.15	8.74	2.16	1.92	39.70	0.044
Agricultural animals	4.19	4.25	3.28	4.15	8.74	2.16	1.92	39.70	0.005
Stray animals	4.12	4.16	2.71	4.15	9.43	2.24	2.07	42.92	0.004
Wildlife	4.24	4.29	2.85	4.28	10.72	2.37	2.38	48.22	0.002
Why care for animals? Indicate how strongly you agree or disagree with the following reasons 1 (Strongly disagree) to 5 (Strongly agree) ^6^ For food safety	4.11	4.04	3.14	3.97	9.55	2.25	2.12	42.96	0.003
For the sake of the animals	3.92	3.75	3.85	3.69	0.70	−0.35	0.56	0.87	0.002
How important are the following conditions in animal care? Indicate how strongly you agree or disagree with the following reasons 1 (Not at all important) to 5 (Very important) ^5^ A comfortable environment	4.22	4.26	3.14	4.30	8.95	2.19	1.93	41.55	0.005
Opportunity to perform natural behaviours	4.11	4.21	3.85	4.15	1.28	0.24	1.02	1.61	0.032
Absences of fear or distress	4.17	4.27	3.14	4.20	6.80	1.90	1.50	30.8	0.013
Absences of pain	4.23	4.28	3.28	4.33	6.38	1.80	1.40	29.01	0.016
Indicate your level of agreement with the following statements 1 (Strongly disagree) to 5 (Strongly agree) ^6^ Procedures performed on animals such as ear tags, castrations and tail docking are acceptable for management	3.53 *	3.29	3.71	3.74	0.71	−0.34	0.57	0.88	0.002
	3.53	3.29	3.71	3.74 *	0.44	−0.8	0.23	0.85	0.014
It is important to have legislation that ensures animal care is adequate	4.26	4.25	3.42	4.33	5.30	1.68	1.16	24.9	0.032

^1^: 1 = Not at all, 2 = Slightly, 3 = Moderately, 4 = Very much, 5 = To a great extent/Extremely. ^2^: 1 = 5%, 2 = 10%, 3 = 20%, 4 = 50%, 5 = 100%, 6 = >100%. ^3^: 1 = Very poor, 2 = Poor, 3 = Satisfactory, 4 = Good, 5 = Very good. ^4^: 1 = Much worse, 2 = Somewhat worse, 3 = About the same, 4 = Better, 5 = Much Better. ^5^: 1 = Not at all important, 2 = Slightly important, 3 = Neither important nor unimportant, 4 = Somewhat important, 5 = Very important. ^6^: 1 = Strongly disagree, 2 = Disagree, 3 = Neither agree nor disagree, 4 = Agree, 5 = Strongly agree. Coef. = coefficient. * Significantly different from those without asterisks.

**Table 3 animals-12-01367-t003:** Age effects on respondents’ perception of attitudes towards animal welfare in China. Mean responses on the binary, 3-point or 5-point ordinal scales are provided. Differences are presented between the reference age group 18–24 years, n = 430, compared with other groups, age 25–34 years (n = 337), 35–44 years (n = 252), 45–54 years (n = 176), 55–64 years (n = 65) and ≥65 years (n = 27).

Questions and Responses	18–24 Years Group Mean score	Comparison Age Group	Mean Scores	Odds Ratio	Coef.	95% CIs Lower Upper		*p*-Value
Have you heard of the phrase ‘animal welfare’?	1.68	≥65	1.84	0.27	−1.29	0.11	0.68	0.005
1 (Never) to 3 (Many times) ^1^								
Do you live in harmony with animals? 1 (Not at all) to 5 (Extremely) ^2^	3.63	25–34	3.26	1.88	0.63	1.4	2.53	<0.0001
		35–44	3.08	2.24	0.8	1.59	3.15	<0.0001
		45–54	3.05	2.08	0.73	1.41	3.06	<0.0001
How important is caring for animals to you as a person? 1 (Not at all) to 5 (Extremely) ^2^	3.76	25–34	3.39	1.77	0.56	1.31	2.39	<0.0001
		35–44	3.33	1.84	0.6	1.3	2.59	0.001
		45–54	3.31	1.62	0.48	1.09	3.39	0.016
Where did you learn about caring for animals? Have not heard (1 = yes, 2 = no)	1.04	25–34	1.06	1.32	0.07	1.13	1.54	<0.0001
		35–44	1.08					
		45–54	1.15					
		55–64	1.16					
		≥65	1.07					
Do you think that animal care should be taught in schools? 1 (Definitely not) to 5 (Definitely) ^3^	3.27	≥65	3.59	0.45	−0.79	0.21	1	0.051
Would you be willing to pay more for products from animals that are better cared for? (1 = yes, 2 = no)	1.33	25–34	1.14	1.14	0.13	1.04	1.25	0.005
		35–44	1.46					
		45–54	1.46					
		55–64	1.61					
		≥65	1.37					
Who do you think is most responsible for the adequate care of animals? All of society	4.12	45–54	3.98	0.61	−0.48	0.4	0.93	0.023
		55–64	4.12	0.5	−0.69	0.29	0.88	0.015
How important is it that the following animals are cared for? 1 (Not at all important) to 5 (Very important) ^4^ Reptiles	4.12	35–44	3.82	1.7	0.53	1.19	2.42	0.003
		≥65	4.22	0.34	−1.08	0.14	0.79	0.013
Insects	3.93	35–44	3.69	1.51	0.41	1.07	2.14	0.018
		≥65	3.88	0.39	−0.93	0.39	0.89	0.025
Pet animals	4.31	35–44	3.99	2.1	0.73	1.46	3	<0.0001
		45–54	4.06	1.68	0.52	1.12	2.53	0.012
Why do people take care of farm animals? Indicate how strongly you agree or disagree with the following reasons 1 (Strongly disagree) to 5 (Strongly agree) ^5^ It makes me feel good	3.97	25–34	4.06	0.73	−0.30	0.54	1	0.049
My religion tells me to	3.51	25–34	3.72	0.63	−0.45	0.47	0.85	0.003
		35–44	3.76	0.58	−0.55	0.41	0.81	0.001
		45–54	3.79	0.58	−0.54	0.39	0.85	0.005
For the sake of the animals	3.78	25–34	3.84	0.72	−0.33	0.53	0.97	0.028
To improve profit from animals	3.88	25–34	3.96	0.56	−0.57	0.41	0.76	<0.0001
		45–54	3.97	0.64	−0.44	0.43	0.95	0.026
		55–64	4.09	0.46	−0.78	0.26	0.8	0.006
How important are the following conditions in animal care? 1 (Not at all important) to 5 (Very important) ^4^ Species-relevant nutrition	4.28	35–44	3.96	1.54	0.43	1.07	2.22	0.019
Control over their environment	4.3	35–44	4.11	1.46	0.37	1.07	1.99	0.018
Absence of pain	4.34	35–44	4.09	1.49	0.39	1.04	2.13	0.028
Indicate your level of agreement with the following statements 1 (Strongly disagree) to 5 (Strongly agree) ^5^ Animals on farm should be provided with enjoyable experiences	4.2	≥65	4.33	0.42	−0.86	0.18	0.98	0.046
It is OK to buy products of animals that have suffered it the price is low enough	3.07	35–44	3.32	0.66	−0.41	0.47	0.92	0.015

^1^: 1 = Never, 2 = A few times, 3 = Many times. ^2^: 1 = Not at all, 2 = Slightly, 3 = Moderately, 4 = Very much, 5 = To a great extent/Extremely. ^3^: 1 = Definitely not, 2 = Probably not, 3 = Possibly, 4 = Probably, 5 = Definitely. ^4^: 1 = Not at all important, 2 = Slightly important, 3 = Neither important nor unimportant, 4 = Somewhat important, 5 = Very important. ^5^: 1 = Strongly disagree, 2 = Disagree, 3 = Neither agree nor disagree, 4 = Agree, 5 = Strongly agree. Coef. = coefficient.

## Data Availability

The raw data have not been published or stored elsewhere but are available on request from FC.

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
