# Peer review of "Gender and Age Effects on Public Attitudes to, and Knowledge of, Animal Welfare in China"

_animals, 2022, doi:10.3390/ani12111367_

Round 1

Reviewer 1 Report

The overall findings of this manuscript are interesting and important.  The authors have clearly carried out a careful and exhaustive analysis of the data.  The data and analysis appear to be sound and I have no concerns about this. I feel that there are some serious deficiencies in the written report. I recommend that authors step back and ensure they have clearly expressed what the data shows in broad strokes before going into the more detailed aspects they find more interesting. I recommend carefully ensuring that the introduction, results, and discussion emphasise the same points and develop them from hypothesis to findings to interpretation. I recommend clearly representing negative findings.  Not only are negative findings important, but they also provide a basis for understanding the significant results and their interactions rather than placing complete trust in the statistical analysis. Alternatively, some other strategy to provide complete/contextualized data presentation before zooming in on elements of interest. I feel that in diving very deep into their data the authors have unfortunately failed to provide a good guide to what the data has to offer the broad readership of this journal who, while guided by the authors, also wish to benefit from understanding the findings as a whole and potentially question some of the authors preferred interpretations. Exactly how this is done is in the hands of the authors but my more specific suggestions are as follows. The introduction emphasizes that a lack of gender effects on animal welfare attitudes in China exists and may be due to gender inequity.  The new data does not specifically assess gender inequity either directly or indirectly.  The Authors conclude there are gender differences mediated by age.  There is a lack of clarity as the authors appear to not find a main effect of gender so this is not really a contradictory finding.  Given the emphasis of the introduction, this issue should be more directly addressed and the equivalence of the current data to the previous studies in China explained to account for whether this study contradicts, or rather extends and develops, their findings. If I have failed to grasp the authors' point here it may reflect the somewhat dense and disjointed way in which the work is presented, almost as if each section had a different author with somewhat different priorities and interests. The abstract and results seem to treat the findings in relation to non-binary and non-disclosed gender as the second (or even primary) finding.  I feel this finding is on very shaky ground with Ns of 7 and 37, respectively. I would suggest phrasing these findings less definitively. The results skip over clearly addressing the hypothesis about gender.  The introduction does not set a very clear hypothesis for the effects of age.  The effects of age and age-gender interactions could be given a more clear overall summary. The results seem to overstep into providing implicit interpretations in some places.  Some of the minor suggestions below relate to this (list NS findings, provide N for all findings). The discussion gives considerable space to other factors that may be important to the existence and expression of animal welfare in Chain (e.g. Confucianism).  Obviously, China is vast, diverse, and culturally should not be expected to conform to many of the norms of animal welfare as it is studied and presented in Europe, America, Australasia etc.  However, this is not consistent with the introduction which conceptualizes having animal welfare concerns largely as the result of knowledge from education.  It is also not consistent with the results section where culturally relevant variables such as religion, urban/rural, field of work, and geographical location receive little or no attention--leaving me to assume their impact on the data is not meaningful or important. Minor points: Pg2, line 65: As the reason for the difference is observational at the inter-country level rather than addressed in the data, more caution should be used in stating it as established fact.  e.g. in the abstract states gender effects vary between counties and may relate to gender equity. At least mention other possible causes. Pg2, line 95: I would suggest that it is worth at least acknowledging that the effects of education are also cultural rather than purely informational/"knowledge". Pg 3, line 18: I think it is important to add here (briefly) the key questions/question types used to assess gender differences in reference 12 and 13 (versus those used in the other references -- and whether this study used something sufficiently similar for direct comparison. Pg 3, Line 122: somewhere add a statement of limitation (e.g. in language choice)  in accessing some minorities, and the effect of cultural diversity within China and other nations would be appreciated.  Understanding that this is not different from what is done in US research using English. Also if the bilingual translators were not coauthors please indicate how they were identified and contacted. This is not a criticism, as having translators familiar with the content is vital, but for clarity. Pg 4, Line 180:   Referring to these seven people as a presumably representative "group" seems highly questionable. Pg 5, line 214: Please provide an N for all groups where they are referenced, e.g. Women aged 18-24 Page 5, line 190.   I feel that it is important to start with a plain statement of the main effects of gender and age.   A data table inclusive of NS may be very large but I feel that that is the necessary context for the significant findings.  Statistical significance is very important but only part of assessing whether a finding is non-random and/or meaningful. Pg 5, line 217 and thereabouts: There are statements without a stated/clear empirical basis, please place them in text rather than requiring the reader to search for them. Pg 10, line 298 -- what specific difference did these studies assess, and what would be the closest equivalent in the current study? Pg . 17, line 524 typo

Author Response

Cover letter: Responses to the reviewers' comments.

Thank you very much to the reviewers for their prompt and valuable comments.

Please note that there was an error in Jin Xiao’s name. First name Jin, family name Xiao. It was correct in the version that I sent.

Added Chair of Nutrition in Estonian affiliation.

All changes and corrections are in the manuscript files and the comments below.

Responses to the 1 reviewer

  • I feel that there are some serious deficiencies in the written report. I recommend that authors step back and ensure they have clearly expressed what the data shows in broad strokes before going into the more detailed aspects they find more interesting. I recommend carefully ensuring that the introduction, results, and discussion emphasise the same points and develop them from hypothesis to findings to interpretation.

We did discuss age and gender in the introduction. We have changed some text in the light of the above, but we think that the introduction, results and discussion (and conclusion too) are focused on the gender and age issues and do not wander into other topics.

  • I recommend clearly representing negative findings. Not only are negative findings important, but they also provide a basis for understanding the significant results and their interactions rather than placing complete trust in the statistical analysis.

The text has been added to present non-significant findings if this is what was meant by ‘negative findings’. We emphasize where there was commonality between age and gender groups. Hopefully, this helps to answer your concerns.

These edits can be found on lines 263 to 277.  “Most of the respondents of all the gender groups declared that they had never heard of animal welfare, but they thought it was extremely essential to care for animals, and perhaps having the possibility to learn about this topic at school would be beneficial. All gender categories felt society was responsible for animal welfare. Pets, mammals, and birds were the animal taxa on which all genders agreed on their importance. For the sake of the environment and human health, the majority of all gender categories considered it important to care for animals. All respondents indicated they care about animals for religious reasons and to make themselves feel better. Increasing profits and increasing the quality of animal products were similarly important to the respondents, regardless of their gender. Irrespective of the gender of the respondents, it was thought important that animals be able to exercise for physical fitness and have control over the environment in which they live. It was also thought to be crucial to have enough area and access to drinkable water and prevent contracting diseases or injury. The management of animals was declared important by the majority of all gender categories, especially concerning transportation time and slaughter management, and they all thought it extremely important to have an animal protection organization.”

On page 9, and lines 311 to 323. We added: “Some factors were consistent across age groups. The age of respondents had no effect on their judgment of animal welfare in China as "poor" and, compared to other countries, they agreed that it was "somewhat worse". In all taxa animals listed in this survey, mammals, birds, and stray animals were identified by all respondents of any age as important, while wild animals, experimental and agricultural animals were identified as extremely important. Regardless of age, all respondents agreed that the most important reasons for caring for animals were food safety, human health, improving the quality of animal products, and the need for animals to be free of fear and allowed to exhibit normal behaviour. The reasons given for the care for animals are that the animals need to have the possibility for physical fitness and control over their environment. Animal care and management were found to be very important regardless of respondent ages, particularly in terms of transit time and slaughter management, but it was also thought highly necessary to have an animal protection organization and law that protects animals.”

 We added in line 191 pg 4: “For clarity, only significant relationships are reported. If response variables are not included it should be understood that there were no significant differences. A complete table with non-significant results is included in Appendix Box A2 and A3.”

  • Alternatively, some other strategy to provide complete/contextualized data presentation before zooming in on elements of interest. I feel that in diving very deep into their data the authors have unfortunately failed to provide a good guide to what the data has to offer the broad readership of this journal who, while guided by the authors, also wish to benefit from understanding the findings as a whole and potentially question some of the authors preferred interpretations.

General overviews are provided at the beginning of the sections. We have tried to add such text to the existing overviews.

  • Exactly how this is done is in the hands of the authors but my more specific suggestions are as follows. The introduction emphasizes that a lack of gender effects on animal welfare attitudes in China exists and may be due to gender inequity. The new data does not specifically assess gender inequity either directly or indirectly.  The Authors conclude there are gender differences mediated by age.  There is a lack of clarity as the authors appear to not find a main effect of gender so this is not really a contradictory finding. 

We have now removed text about gender inequity.

Lines 16 page 1, We deleted “and households where the women are empowered to make their own decisions.

Line 24 page 1, We deleted “The latter generally indicated less positive attitudes towards animals and their management, and those who chose not to reveal their gender indicating more positive attitudes.

Line 28 page 1, We deleted “who are more gender empowered. ”

And we have provided some general findings text at the start of the sections. We consider the discovery of a gender x age interaction to be an elaboration of existing knowledge, not a contradiction.

  • Given the emphasis of the introduction, this issue should be more directly addressed and the equivalence of the current data to the previous studies in China explained to account for whether this study contradicts, or rather extends and develops, their findings. If I have failed to grasp the authors' point here it may reflect the somewhat dense and disjointed way in which the work is presented, almost as if each section had a different author with somewhat different priorities and interests. The abstract and results seem to treat the findings in relation to non-binary and non-disclosed gender as the second (or even primary) finding. I feel this finding is on very shaky ground with Ns of 7 and 37, respectively. I would suggest phrasing these findings less definitively.

We have deleted text in the abstract that might seem to be overreaching in this regard.

Line 41 page 1 , We deleted “Non-binary participants generally indicated less positive attitudes towards animals and their management, and those who chose not to reveal their gender indicating more positive attitudes.”

Line 233 page 7, We deleted: “Non-revealers also reported that they would be willing to pay more than 5 % for a product from an animal that had been very well cared for, compared to men and women, who did not report, on average, that they would pay more.”

We Added in line 237 page 7: “although the number of such respondents is recognized as being too small for definitive comment.”

We added in line 358 page 11:” While the number of non-binary declaring respondents was small, they are included as their responses are entirely novel and,  despite their small number, suggest that there may be differences in attitudes to the mainstream male and female identities.”

 The results include justification of our inclusion of non-binary respondents despite their small numbers and we have not definitively emphasised this. We would contend that the number of non-revealers at 39 is sufficient to warrant comment and comparison. The statistical analysis method used is robust against unequal numbers but the concern with small numbers is that they do not accurately represent the population that they come from. Many studies are conducted with just 39 respondents, so we consider this reasonably reliable, and given the novelty of this differentiation of gender responses, worthy of reporting. Regarding the seven respondents, we do not emphasize this at any point in the reporting of results, or discussion, as we consider this to be too small a sample for representativeness.

  • The results skip over clearly addressing the hypothesis about gender. The introduction does not set a very clear hypothesis for the effects of age. 

Text has been added to clarify this on lines 113 to 115 and page 3.”It was hypothesized that among the general public in China attitudes to animals would differ with gender, and that attitudes to animals would differ with age.”

We added in line 82 page 2:” Much of this previous research on attitudes of Chinese people focused on students as the respondents. However, it is important to further understand the attitudes of society at large, including across all age groups.”

  • The results seem to overstep into providing implicit interpretations in some places. Some of the minor suggestions below relate to this (list NS findings, provide N for all findings). The discussion gives considerable space to other factors that may be important to the existence and expression of animal welfare in Chain (e.g. Confucianism).  Obviously, China is vast, diverse, and culturally should not be expected to conform to many of the norms of animal welfare as it is studied and presented in Europe, America, Australasia etc.  However, this is not consistent with the introduction which conceptualizes having animal welfare concerns largely as the result of knowledge from education.

Education is only discussed in relation to explaining the age differences, not a variable to be considered separately. We do not agree that the introduction conceptualizes having animal welfare concerns largely as the result of knowledge from education. We have responded to your minor comments as below.

  • It is also not consistent with the results section where culturally relevant variables such as religion, urban/rural, field of work, and geographical location receive little or no attention--leaving me to assume their impact on the data is not meaningful or important.

These variables have now been removed from the text. Lines 169 page 4 We deleted: “religion, high education level, work field, and living place).” We wanted to focus on age and gender in this paper because these clearly both had major effects on attitudes.

Minor points:

  • Pg2, line 65: As the reason for the difference is observational at the inter-country level rather than addressed in the data, more caution should be used in stating it as established fact. g. in the abstract states gender effects vary between counties and may relate to gender equity. At least mention other possible causes.

 Yes, we have now deleted the following which may be indeed be considered incautious as we did not actually test this:“Some of these global differences between the attitudes of women and men towards animals may be explained by the level of dependence of women on men.” Add: “Global differences in the attitudes of women and men towards animals may be ex-plained by a variety of factors, such as political and cultural influences and the level of dependence of women on men”

  • Pg2, line 95: I would suggest that it is worth at least acknowledging that the effects of education are also cultural rather than purely informational/"knowledge".

Pg2, line 95: Added in the text as suggested:” For example Davey, 2006, suggests that young people in China have tended to have better educational opportunities, including different cultural education, and have gained more knowledge relating to animal welfare than older people [25].”

  • Pg 3, line 18: I think it is important to add here (briefly) the key questions/question types used to assess gender differences in reference 12 and 13 (versus those used in the other references -- and whether this study used something sufficiently similar for direct comparison.

 Pg 2, line 58: Added in the text: “In contrast, however, previous studies in the People’s Republic of China (hereafter China) [12,13] found similarities between women and men regarding their attitudes towards animals. However, Li et al. [12] focused only on the transport and slaughter of animals, and Su and Martens [13] reported that women and men answered in a similar way to general questions regarding their attitudes to animals.”

  • Pg 3, Line 122: somewhere add a statement of limitation (e.g. in language choice) in accessing some minorities, and the effect of cultural diversity within China and other nations would be appreciated.  Understanding that this is not different from what is done in US research using English. Also if the bilingual translators were not coauthors please indicate how they were identified and contacted. This is not a criticism, as having translators familiar with the content is vital, but for clarity.
  • Pg 4, Line 180: Referring to these seven people as a presumably representative "group" seems highly questionable.

Pg 4 line 179 changed in “gender categories

  • Pg 5, line 214: Please provide an N for all groups where they are referenced, e.g. Women aged 18-24

Pg 5 line from 199 to 210: Added in the text N for all groups as suggested.

  • Page 5, line 190. I feel that it is important to start with a plain statement of the main effects of gender and age.   A data table inclusive of NS may be very large but I feel that that is the necessary context for the significant findings.  Statistical significance is very important but only part of assessing whether a finding is non-random and/or meaningful.

Pg 5 line 199 “Women that were younger did advance more animal welfare opinions than the other groups.

We have now added appendices 2 and 3 with the non-significant findings. We have also added some text to lines 263 to 277 on page 9, and lines 311 to 323 as above described.  

  • Pg 5, line 217 and thereabouts: There are statements without a stated/clear empirical basis, please place them in text rather than requiring the reader to search for them.

Pg 7. Line 229 We have added: “Of non- revealers, 60% were prepared to pay more (female 61% and male 56%). Most of the men (60%) and women (65%) were prepared to pay at least 10%, while for non-revealers, most (70%) were prepared to pay 20% more for a product from an animal that had been very well cared for.”

  • Pg 10, line 298 -- what specific difference did these studies assess, and what would be the closest equivalent in the current study?

Added in the text as suggested

  • Pg . 17, line 524 typo

Pg 17 line 424 old version?..., We could not find any typo but have screened the manuscript carefully for any typos or grammatical errors.

Reviewer 2 Report

While the article was well written and made use of existing data, I had trouble identifying any practical reason to pursue this topic. I never learned of any real application for the analysis completed here. It seemed that this project was an attempt to wring a few more drops of "publication credit" from data that had already been adequately explored for the most important results. The results were so scattered that it was very hard to follow which groups and subgroups were being discussed at any time.  The tables are too busy and challenging to understand without really working hard to pick out the important pieces. That said, I do feel that this might be useful if the authors are able to clearly identify and connect a practical use of the data they developed from their analysis. In addition, the very small numbers of "non-binary" and "not revealing" responses make the generalizations in this article a bit far-fetched.  I think there may be something interesting there, but they need many more respondents before this can be identified.

I hope the authors take some time to research the real-world applications of their results.  With that information added and discussed,  this would be a nice article.

Author Response

Thank you very much to the reviewers for their prompt and valuable comments.

Please note that there was an error in Jin Xiao’s name. First name Jin, family name Xiao. It was correct in the version that I sent.

Added Chair of Nutrition in Estonian affiliation.

All changes and corrections are in the manuscript files and the comments below.

Responses to the 2 reviewer

  • While the article was well written and made use of existing data, I had trouble identifying any practical reason to pursue this topic. I never learned of any real application for the analysis completed here. It seemed that this project was an attempt to wring a few more drops of "publication credit" from data that had already been adequately explored for the most important results. The results were so scattered that it was very hard to follow which groups and subgroups were being discussed at any time.  The tables are too busy and challenging to understand without really working hard to pick out the important pieces.

We do not know of another way of presenting these results and still presenting the necessary statistical information. It is a standard method for attitudinal studies of this nature. We have added a full set of results, including non-significant results, in response to a comment by reviewer 1.

  • That said, I do feel that this might be useful if the authors are able to clearly identify and connect a practical use of the data they developed from their analysis. In addition, the very small numbers of "non-binary" and "not revealing" responses make the generalizations in this article a bit far-fetched.  I think there may be something interesting there, but they need many more respondents before this can be identified.

We would contend that the number of non-revealers at 39 is sufficient to warrant comment and comparison. The statistical analytical method used is robust against unequal numbers but the concern with small numbers is that they do not accurately represent the population that they come from. Many studies are conducted with just 39 respondents, so we consider this reasonably reliable, and given the novelty of this differentiation of gender responses, worthy of reporting. Nevertheless, we have removed the results for these groups from the Abstract and Simple Summary. Regarding the seven respondents, we do not emphasize this at any point in the reporting of results, or discussion, as we consider this to be too small a sample for representativeness.

  • I hope the authors take some time to research the real-world applications of their results.  With that information added and discussed,  this would be a nice article.

The text has now been added to answer these comments. Lines 499 to 508 p. 14.

’’This study provides evidence that the attitudes of the Chinese population and relationships between these attitudes and age and gender are not greatly different from those recorded elsewhere; the sensitivity of the Chinese population to animal welfare does not differ much from the rest of the world. The identification of gender and age groups that are more concerned with the welfare of animals confirms the value of, and interest in, educating the young in the formal school setting on issues of animal welfare. Whether this interest is related to the age of the respondents, and we might expect that their attitudes would harden as they get older, or it is a generational factor, with continued interest into later life, could be confirmed by a longitudinal study, ideally asking the same respondents in later life, but this is beyond the bounds of this study.“

Round 2

Reviewer 2 Report

I appreciate the authors' responses to my concerns, but fail to see any significant changes in clarity or impact. However, it can certainly be published. I think the paragraph that was added (regarding a longitudinal study) provides the most striking potential impact of the project. I'd like to see how something like that turned out- it would be interesting to know if attitudes changed with age or if it's a generational difference.